# Collagen-Specific Molecular Magnetic Resonance Imaging of Prostate Cancer

**DOI:** 10.3390/ijms24010711

**Published:** 2022-12-31

**Authors:** Avan Kader, Jan O. Kaufmann, Dilyana B. Mangarova, Jana Moeckel, Lisa C. Adams, Julia Brangsch, Jennifer L. Heyl, Jing Zhao, Christine Verlemann, Uwe Karst, Federico Collettini, Timo A. Auer, Bernd Hamm, Marcus R. Makowski

**Affiliations:** 1Department of Radiology, Charité—Universitätsmedizin Berlin, Corporate Member of Freie Universität Berlin, Humboldt-Universität zu Berlin, and Berlin Institute of Health, Charitéplatz 1, 10117 Berlin, Germany; 2Department of Biology, Chemistry and Pharmacy, Institute of Biology, Freie Universität Berlin, Königin-Luise-Str. 1-3, 14195 Berlin, Germany; 3Department of Diagnostic and Interventional Radiology, Technical University of Munich, Ismaninger Str. 22, 81675 Munich, Germany; 4Bundesanstalt für Materialforschung und -prüfung (BAM), Division 1.5 Protein Analysis, Richard-Willstätter-Str. 11, 12489 Berlin, Germany; 5Department of Chemistry, Humboldt-Universität zu Berlin, Brook-Taylor-Str. 2, 12489 Berlin, Germany; 6Institute of Inorganic and Analytical Chemistry, University of Münster, Corrensstr. 48, 48149 Münster, Germany; 7BIH Charité Clinician Scientist Program, Berlin Institute of Health at Charité-Universitätsmedizin Berlin, 10178 Berlin, Germany; 8School of Biomedical Engineering and Imaging Sciences, United Kingdom, King’s College London, St Thomas’ Hospital Westminster Bridge Road, London SE1 7EH, UK

**Keywords:** molecular imaging, magnetic resonance imaging, prostate cancer, collagen

## Abstract

Constant interactions between tumor cells and the extracellular matrix (ECM) influence the progression of prostate cancer (PCa). One of the key components of the ECM are collagen fibers, since they are responsible for the tissue stiffness, growth, adhesion, proliferation, migration, invasion/metastasis, cell signaling, and immune recruitment of tumor cells. To explore this molecular marker in the content of PCa, we investigated two different tumor volumes (500 mm^3^ and 1000 mm^3^) of a xenograft mouse model of PCa with molecular magnetic resonance imaging (MRI) using a collagen-specific probe. For in vivo MRI evaluation, T1-weighted sequences before and after probe administration were analyzed. No significant signal difference between the two tumor volumes could be found. However, we detected a significant difference between the signal intensity of the peripheral tumor area and the central area of the tumor, at both 500 mm^3^ (*p* < 0.01, *n* = 16) and at 1000 mm^3^ (*p* < 0.01, *n* = 16). The results of our histologic analyses confirmed the in vivo studies: There was no significant difference in the amount of collagen between the two tumor volumes (*p* > 0.05), but within the tumor, higher collagen expression was observed in the peripheral area compared with the central area of the tumor. Laser ablation with inductively coupled plasma mass spectrometry further confirmed these results. The 1000 mm^3^ tumors contained 2.8 ± 1.0% collagen and the 500 mm^3^ tumors contained 3.2 ± 1.2% (*n* = 16). There was a strong correlation between the in vivo MRI data and the ex vivo histological data (y = −0.068x + 1.1; R^2^ = 0.74) (*n* = 16). The results of elemental analysis by inductively coupled plasma mass spectrometry supported the MRI data (y = 3.82x + 0.56; R^2^ = 0.79; *n* = 7). MRI with the collagen-specific probe in PCa enables differentiation between different tumor areas. This may help to differentiate tumor from healthy tissue, potentially identifying tumor areas with a specific tumor biology.

## 1. Introduction

There is a constant interaction between tumor cells and other cells, promoting cell migration, development, and homeostasis [1]. Collagen fibers are associated with tissue stiffness but also with the growth, adhesion, proliferation, migration, invasion/metastasis, cell signaling, and immune recruitment of tumor cells [2,3,4,5]. Twenty-eight different types of collagen have been classified based on their supramolecular structures [4]. The regulation of collagen fibers is regulated by other proteins and enzymes of the extracellular matrix (ECM), involving collagen expansion, fiber thickness and fiber packing. The formation of collagen niches in some cells is a strategy for influencing nearby cells [6,7,8,9,10]. The remodeling of collagen in tumor diseases leads to pathological characteristics of cancer progression [4]. Fibrillar collagens, especially collagen type I, provide mechanical strength to the ECM and enable resistance to deformation and breakage [11,12,13,14]. They are covalently cross-linked by enzymes from the lysyl oxidase (LOX) family [15]. High LOX activity contributes to increased stiffening of the ECM and, when overexpressed, is associated with metastasis and poor outcome [2,16,17].

Studies have shown that malignant prostate tissue is almost 60% stiffer than benign prostate tissue [18,19,20]. Prostate cancer is characterized by a high degree of heterogeneity, posing a major challenge for non-invasive grading and stratifying patients for treatment allocation, especially for precision therapy [21]. An increase in collagen type I has already been observed in the precursor lesion of PCa, intraepithelial neoplasia, suggesting that collagen content plays a role in the development of PCa at an early stage of development [22,23]. There is a close relationship between collagen-related tumor stiffness and tumor development and aggressiveness in PCa [14]. That is, when the collagen fibers are more aligned, the matrix is stiffer and the PCa becomes more aggressive. [18,19,20,24,25]. The stiffness of the tissue produced by collagen is a good parameter for the detection of prostate tumors, but linearization of collagen can also be a helpful tool for detection. ECM stiffening in PCa is a critical driver of tumor growth, invasion, and metastasis [26,27].

Molecular magnetic resonance (MR) imaging has been the focus of intense research for some time, as it can visualize biological processes at the cellular and subcellular level without invasive procedures. The detection and localization of molecular markers of disease, such as cell surface receptors, enzymes, and the ECM, are optimal for molecular imaging. In 2007, Caravan et al. first reported a molecular contrast agent in MRI that binds to collagen [28]. The collagen-specific contrast agent (EP-3533) consists of a cyclic peptide of 16 amino acids with three Gd-DOTA chelates and has already been successfully tested in various diseases, including myocardial infarction [28], liver fibrosis [29], abdominal aortic aneurysm [30], and tumorous diseases such as pancreatic cancer [31]. It shows saturated binding to collagen type I. Here, we investigate the feasibility and applicability of a collagen-specific probe for MR imaging and examine its use in PCa with different sizes.

## 2. Results

In this study, mice with PCa were administered a collagen-specific probe to visualize collagen fibers in vivo by MRI. Two tumor sizes, 500 mm^3^ and 1000 mm^3^, were studied and compared. Figure 1 shows the study design.

All animals developed a tumor (*n* = 32), but there were differences in tumor development, although the number of cells and the timeframe of the study were standardized. Tumor size was measured and documented daily using a caliper. Final tumor size of 500 mm^3^ was achieved between 53 and 149 days after surgery, and tumor size of 1000 mm^3^ was achieved between 43 and 113 days after surgery.

### 2.1. Characterization in T1-Weighted MR Imaging Using a Collagen-Specific Probe

After a plain mouse MRI scan, intravenous administration of 10 µmol/kg collagen-specific probe was performed via the tail vein. The difference between the pre- and post-contrast images (Figure 2a,b) showed a clear increase in signal after administration of the collagen-specific probe.

A clear differentiation between tumorous and healthy tissue was possible. The difference between the pre- and post-contrast images was significant for both tumor sizes (500 mm^3^ tumor, *p* < 0.01; 1000 mm^3^ tumor, *p* < 0.01, *n* = 16 for each tumor). For tumor size 1000 mm^3^, an SI of 800.5 ± 103.6 was determined before probe administration, and 1482.0 ± 156.0 after probe administration, with a RE of 0.81 ± 0.2 (*n* = 16). For tumor size 500 mm^3^, SI was determined to be 916.7 ± 210.6 before probe administration and 1590.9 ± 295.7 after probe administration, RE of 0.72 ± 0.3 (*n* = 16). No significant difference was detected between 500 mm^3^ and 1000 mm^3^ tumors post contrast (*p* = 0.13).

In addition, the peripheral area and the central area of the tumor were compared (Figure 2f). Here, we could find a significant difference within each tumor, regardless of tumor size. For 1000 mm^3^ tumors, the RE was 0.8 ± 0.1 (*n* = 16) in the central area and 1.2 ± 0.3 (*n* = 16) in the peripheral area. The differences were significant, with *p* < 0.01. For 500 mm^3^ tumors, the RE was 0.8 ± 0.1 in the central area (*p* < 0.01).

### 2.2. Ex Vivo Collagen Analysis

In both tumor sizes, collagen fibers were detected in the tissue using PSR stain (Figure 3b,c). The collagen fibers are shown in red–yellow–green. Figure 3a shows an H&E stain for the tumor overview.

To determine the collagen content, three different areas were selected for each probe and the percentage of the collagen presence was determined using the analyzer.

The analysis showed no significant difference in collagen content between the two tumor sizes. The 500 mm^3^ tumors had an average of 3.2 ± 1.2% collagen and the 1000 mm^3^ tumors showed 2.8 ± 1.0% collagen (Figure 4a).

Overall, an increased presence of collagen fibers could be observed, especially in the peripheral area (Figure 4b). The 1000 mm^3^ tumors contained 3.8 ± 1.6% collagen in the peripheral area and 1.5 ± 0.6% collagen in the central area (*n* = 16, *p* < 0.001). The analysis of the 500 mm^3^ shows 4.4 ± 1.6% in the peripheral area and 2.1 ± 1.0% in the central area (*n* = 16, *p* < 0.001). Neither the peripheral (*p* = 0.2) nor the central areas (*p* = 0.07) demonstrated a significant difference in collagen content when comparing the 1000 mm^3^ and 500 mm^3^ tumors.

To confirm the results, a total collagen assay was conducted (Figure 4c), which shows no significant difference between both tumor volumes, for 1000 mm^3^ 0.2 ± 0.03 RFU and for 500 mm^3^ 0.2 ± 0.04 RFU, *p* = 1.0 (*n* = 3 for each tumor size).

The in vivo T1-weighted MRI measurements were correlated with the ex vivo data from collagen PSR staining. A strong correlation was found (y = −0.0683x + 1.10; R^2^ = 0.74) (Figure 5a).

### 2.3. Elemental Analysis of Tumor Tissue with Specific Regard to Gd

LA-ICP-MS analysis was performed to localize Gd in the tumor tissue. For each tumor size, *n* = 3 mice were examined. The LA-ICP-MS data showed a good colocalization of the Gd with the histological data, as shown in Figure 3c. Gd could be detected in the peripheral area more than in the intra-tumoral space.

To accurately determine the concentration of Gd in the tissue, a quantitative ICP-MS analysis was conducted. The determined Gd content in the tumor tissue was correlated with the MRI RE data, which showed a good correlation (y = 3.82x + 0.56; R^2^ = 0.79) (Figure 5b). For each group, *n* = 7 measurements were performed.

## 3. Discussion

In this study, we demonstrated a collagen-specific probe allows the MR-based characterization of PCa in an LNCaP tumor mouse model. After administration of the collagen-specific probe, a higher signal intensity was observed compared to pre-contrast imaging. The contrast agent allowed for a clear differentiation between tumor tissue and healthy tissue. Comparing two different tumor volumes (500 mm^3^ and 1000 mm^3^), no significant difference in MRI signal intensity was observed. This was also confirmed by ex vivo analyses. Regardless of tumor size, more collagen fibers were found in the peripheral than in the central tumor region.

In view of the biological processes involved in the development of PCa, specific markers can be used for molecular imaging on MRI. PET probes are mainly used for the molecular imaging of PCa, e.g., choline [32] and acetate [33]. However, PET analogues are associated with limitations including their cost and logistics. Additionally, the spatial resolution of PET is limited compared to MRI. Therefore, small-molecule MRI probes represent an ideal alternative for the detection of specific disease markers, overexpressed molecules, and receptors [34]. The tumor microenvironment, which includes tumor vessels, connective tissue, infiltrated immune cells, and the extracellular matrix, plays a crucial role in tumor development. It is essential for progression and metastasis [35,36,37]. The ECM contains collagen as one of its major components. The connection between molecular imaging and collagen has already been demonstrated for other pathologies, e.g., cardiovascular [30] or fibrosis-related diseases [28]. In this study, we were able to show that prostate carcinomas can be visualized and characterized with the collagen-specifical contrast agent in MRI. Contrast-enhanced MR images allowed for a clear delineation of the tumor from surrounding healthy tissue.

Other molecular probes such as elastin-specific probes and feraheme have shown the feasibility and differentiation between different tumor sizes in terms of elastin content and/or macrophage presence [38] in previous studies [39]. This observation could not be made for the collagen-specific contrast agent. Both tumor sizes, 500 mm^3^, and 1000 mm^3^ did not show a significant difference in collagen content as measured by MRI examination and histology. This shows that collagen is expressed independently of tumor size and can be a good tool for detecting PCa in both early diagnosis and advanced PCa. Collagen synthesis and maturation is a complex process, involving increased transcription and translation of collagen coding genes and upregulation of enzymes. Collagen influences cancer cell behavior, but cancer cells also shape collagen to form a reinforcing cell collagen loop that promotes tumor progression [40].

Since prostate tumor tissue differs from normal prostate tissue [41], a change in stiffness may indicate abnormality. Collagen deposition and linearization are strongly related to macrophage activity [42]. A study by Acerbi et al. examined the ECM in breast cancer progression, finding that breast cancer progression and aggression are related to collagen linearization and stromal stiffening [42]. Tumor collagen internalization is mediated by tumor-associated macrophages (TAMs) and cancer-associated fibroblasts (CAFs), with M2-like TAMs being the dominant collagen-internalizing cell type [43,44]. Macrophages and collagen play an important role in disease development and tumor microenvironment [45], so a combination of the collagen-specific probe and an iron oxide probe, in which iron is taken up by macrophages, would be a way to obtain more information about the tumor. A previous study by Adams et al. demonstrated the applicability of the collagen-specific probe in combination with an iron oxide probe in abdominal aortic aneurysm [30].

Regardless of the size of the tumors, we found that collagen fibers were more abundant in the peripheral area than in the rest of the tumor. The alignment of the collagen fibers is important for the remodeling of the ECM caused by tumor cells. Tumor growth results in strain, which causes the collagen fibers to peripherally align with the tumor circumference. The linearization of collagen fibers provides an opportunity for direct invasive growth of cancer cells. [46,47,48,49]. The tension formed in the orientation of the collagen fiber bundles increases the stiffness of the tumor [50]. The crucial mechanism of orientation in malignant tumors is still unclear, but it is known that tumor cells, CAFs, mesenchymal cells, and interstitial fluid are involved in the process. Further studies are needed to understand the full process of collagen fiber orientation in tumor diseases and their respective stages.

This study has limitations: Most importantly, an orthotopic mouse model would be a more optimal tumor model than the xenograft mouse model, so that the tumor microenvironment can also be considered. In addition, triggering PCa in the mouse prostate would better mimic the pathophysiology of human disease. This requires higher-resolution MRI because the mouse prostate is very small. Molecular characterization is only possible for collagen and shows no general difference in tumor size, which is both a limitation and an advantage. However, the entire progression from the origin to progression or metastasis of collagen would need to be analyzed to form more reliable conclusions.

Collagen-specific molecular MRI of PCa enables the differentiation between different tumor areas. This may help to differentiate tumor from healthy tissue, potentially identifying tumor areas with a specific tumor biology.

## 4. Materials and Methods

### 4.1. Cell Culture

LNCaP cells were purchased from ATCC^®^ CRL-1740™ (Manassas, VA, USA) and grown in Roswell Park Memorial Institute (RPMI) 1640 medium (Gibco™, Thermo Fischer Scientific, Waltham, MA, USA) and supplemented with 10% fetal calf serum (FCS) (Gibco™, Thermo Fischer Scientific, Waltham, MA, USA) and 5% penicillin-streptomycin (Gibco™, Thermo Fischer Scientific, Waltham, MA, USA). Cells were cultured in 150 cm^3^ tissue culture flasks until they were about 75–80% confluent. Then, the cells were washed with phosphate-buffered saline (PBS) (Gibco™, Thermo Fischer Scientific, Waltham, MA, USA), trypsinized and resuspended in 1 mL RPMI medium without supplements. Cells were grown at 37 °C and 5% CO_2_.

A 0.4% trypan blue solution (Gibco™, Thermo Fischer Scientific, Waltham, MA, USA) was used to count the cells.

### 4.2. Xenograft Mouse Model

Experiments were performed in accordance with the local guidelines and regulations for the implementation of the Animal Welfare Act and the regulations of the Federation of Laboratory Animal Science Associations (FELASA). Animal experiments were approved by the supervisory authority of the Berlin State Office for Health and Social Affairs (LAGeSo) (G0094/19). Eight-week-old male SCID mice (CB17/Icr-Prkdcscid/IcrIcoCrl) were purchased from Charles River Laboratories (Sulzfeld, Germany) (*n* = 32). The animals were separated into two separate groups (*n* = 16), namely 500 mm^3^ and 1000 mm^3^ tumor size as endpoint.

For mice anesthesia, Medetomidine (500 µg/kg), midazolam (5 mg/kg), and fentanyl (50 µg/kg) were injected intraperitoneally. A cell suspension containing 2 × 10^6^ LNCaP cells was injected subcutaneously (s.c.) into the right scapula area. Anesthesia was antagonized with atipamezole (750 µg/kg), flumazenil (0.5 mg/kg), and naloxone (1200 µg/kg).

The final tumor size was measured with a caliper. When the tumor size of 500 mm^3^ (*n* = 16) or 1000 mm^3^ (*n* = 16), respectively, was reached, MR imaging was performed on the anesthetized animal. In previous studies, it was shown that differences in the composition of the ECM were visible at the sizes of 500 mm^3^ and 1000 mm^3^, so these two sizes were chosen [38,39]. Following MR imaging, the mice were euthanized, and the tumor tissue was taken for ex vivo studies.

### 4.3. In Vivo MRI

In vivo MR imaging was performed using a 3.0 Tesla MR scanner (MAGNETOM Lumina, Siemens, Erlangen, Germany) and a 4-channel receive coil array for mouse body applications (Mouse scapula Array, P-H04LE-030, version1, Rapid Biomedical GmbH, Germany). After induction of i.p. anesthesia, the mice were placed in a prone position on the MRI patient table. A venous port was placed via the tail vein for the administration of the contrast agent during MR imaging. The temperature of the body (37 °C) was monitored with an MR-compatible heating system (model 1025, SA Instruments Inc, Stony Brook, NY, USA) to prevent rapid cooling.

### 4.4. Collagen-Specific Imaging Using T1-Weighted Sequences

For this study, a collagen-specific probe, EP-3533 [28,51] was used. The sample was developed by Caravan et al. [28]. It is a cyclic peptide that specifically binds against collagen type 1. For the measurement, 10 µmol/kg was injected intravenously and was performed at 3.0 Tesla MRI. The mice were placed in prone position after anesthesia and scanned with a 4-channel receiver coil array.

A three-dimensional localizer scan was used to localize the tumor in low resolution, performed in sagittal, coronal, and transverse orientation with the following parameters: field-of-view (FOV) = 280 × 280 mm, matrix = 320, slice thickness = 1.5 mm, repetition time (TR) = 11.0 ms, echo time (TE) = 5.39 ms, flip angle = 20° and number of slices = 10. T2-weighted anatomical images were acquired using the following parameters: FOV = 150 mm, matrix = 201, slice thickness = 1.2 mm, TR = 3200.0 ms, TE = 77.0 ms, flip angle = 140°, and a number of slices = 25. To visualize the collagen-specific probe, a T1-weighted sequence with the following parameters was performed: FOV = 70 mm, matrix = 131, slice thickness = 0.4 mm, TR = 833.8 ms, TE = 6.34 ms, flip angle = 30° and number of slices = 30.

### 4.5. MRI Measurements

Present MR images were evaluated with Visage 7.1 (version 7.1, Visage Imaging, Germany). T1-weighted images were compared before and after administration of the collagen-specific probe and the signal intensity (SI). For the evaluation of relative enhancement (RE), 2D regions of interest (ROIs) were drawn around the corresponding areas in the MR images pre- and post-contrast administration. The central area was defined by means of a midpoint. Starting from this point, a radius was defined, which was measured in each case. For the peripheral area, an outermost point starting from the midpoint was selected, which was measured in each case. The calculation of the relative enhancement (RE) was based on the following formula:(1)$$RE=\frac{\left(SI_{postcontrast}-SI{\;}_{precontrast}\right)}{SI{\;}_{precontrast}}$$

### 4.6. Histological Staining

Frozen tumor probes were cut in 10 μm thick serial sections at −20 °C. For fixation of the sample, acetone (≥99%, Fisher scientific, Hampton, NY, USA) was intubated at −20 °C for 6 min. A picrosirius red stain (PSR) was performed to detect the collagen fibers and Hematoxylin and Eosin (H&E) were performed for the overview.

The sections were analyzed using a Keyence microscope (BZ-x800 Series, Osaka, Japan) and with the help of a polarisation filter (Osaka Prefecture, Osaka, Japan). With the polarisation filter, it was possible to distinguish between different types of collagens.

### 4.7. Quantification of the Collagen by Histological Analysis

The amount of collagen was measured using BZ-X800 Analyzer image analysis software (Keyence, Osaka, Japan). For each sample, three representative areas (two peripheral areas and two central areas) were analyzed. In each case, the mean value was calculated. First, the whole tumor region was marked. Then, all stained collagen fibers were identified and the ratio to the total labeled tumor region was calculated as a percentage based on the labeled pixels.

### 4.8. Quantitative Measurement of Collagen

For an additional tumor collagen analysis, the QuickZyme Bioscienced collagen assay (QuickZyme Biosciences B.V., Leiden, The Netherlands) was used. First, a piece of the tumor was cut and dried. For the dry tumor samples (*n* = 3 per group), 6M HCl was added to each tube and incubated for 20 h at 95 °C in a thermoblock (eppendorf, Hamburg, Germany). After incubation, the tubes were cooled to room temperature and centrifuged for 10 min at 13,000 rpm. For further analysis, 35 μL of the supernatant was used. The hydrolyzed samples were diluted with demineralized water: 1 volume sample was added with 0.5 volume demineralized water and standard controllers were included. Then the assay buffer was added and incubated for 20 min at room temperature while shaking. The detection reagent sufficient was prepared by mixing detection reagents A and B (2:3) and, for each well, 75 μL of the detection reagent was added. The plate was covered and shaken, followed by an incubation time of 60 min at 60 °C, and then cooled down to room temperature for 5 min on ice. Last, the plate was mixed, and the absorbance was measured by 570 nm by using the photometer (PowerWave HT Microplate Spectrophotometer, BioTek Instruments, Winooski, VT, USA).

### 4.9. Laser Ablation–Inductively Coupled Plasma–Mass Spectroscopy (LA-ICP-MS)

The tumor samples were cut into 10 µm cryosections at −20 °C and mounted on SuperFrost Plus slides (Thermo Fisher Scientific, Waltham, MA, USA). Localization of gadolinium in the thin section was performed by laser ablation–inductively coupled plasma–mass spectrometry (LA-ICP-MS) (*n* = 3 per group) and analyzed with in-house-developed software Imajar 3.64 (written by Robin Schmid). For this purpose, the imageBIO266 laser ablation system (Elemental Scientific Lasers, Bozeman, MT, USA) equipped with a two volume ablation cell (Elemental Scientific Lasers) was coupled to an iCAP triple quad (TQ)-ICP-MS (Thermo Fisher Scientific) via a dual concentric injector (DCI, Elemental Scientific Lasers). The samples were ablated with a laser repetition rate of 100 Hz and a laser fluence of 2.6 J/cm^2^ in a line-by-line approach with a spot size of 20 µm and a scan speed of 200 µm/s and transported to the ICP-MS with a He gas flow of 1400 L/min. The ICP-MS was equipped with a nickel interface and the plasma power was 1550 watt. Argon was used as cool gas (flow rate 14 L/min), as auxiliary gas (flow rate 0.8 L/min) and nebulizer gas (flow rate was tuned daily). The TQ-MS was operated with oxygen as reaction gas and the ions recorded were ^158^Gd^16^O^+^, ^56^Fe^16^O^+^, ^66^Zn^+^, ^65^Cu^+^ and ^31^P^16^O^+^ with a dwell time of 20 ms each. To determine the Gd concentration, an external calibration with matrix-matched gelatin standards was used. For that, 100 mg gelatin (Carl Roth, Karlsruhe, Germany) was mixed with 900 µL solution of gadolinium chloride (Alfa Aesar, Haverhill, MA, USA) in a concentration range of 0–1 mg and heated up to 80 °C. After homogenization with the vortex mixer Vortex-Genie 2 (Scientific Industries, Bohemia, NY, USA) a droplet of the mixture was frozen at −20°C and cut into 10 µm thin sections with the Cryostar NX70 cryostat (Thermo Fisher Scientific). Eleven lines with a length of 2 mm each were ablated using the previously described method. After discarding the first line of each standard, the averaged intensities were plotted against the concentration of the gelatin standards and analyzed using weighted linear regression [52]. A LOD of 13 ng/g and a LOQ of 44 ng/g were calculated as three or ten times the standard deviation divided by the slope of the regression, respectively. The concentrations of the gelatin standards were analyzed after digestion with HNO_3_ 65 % (VWR Chemicals, Radnor, PA, USA) by ICP-MS.

### 4.10. Inductively Coupled Plasma-Mass Spectrometry (ICP-MS)

For the determination of the total Gadolinium (Gd) concentration in the tumor samples, a cross-section of the tumor probe was prepared (*n* = 7 per group) and then dehydrated under a vacuum atmosphere (vacuum pumping device, vacuubrand, Wertheim, Germany). For digestion 750 µL of 68 % HNO_3_ (Thermo Fisher Scientific) was added to each sample. A total of 500 µL of the digested samples was diluted with 9.45 mL MQ H_2_O. For the external calibration, dilutions of Gd-ICP-MS-stock solution (Merck, Darmstadt, Germany) in a concentration range of 0–15 ng/g acidified with 500 µL HNO_3_ (VWR Chemicals, Radnor, PA, USA) was used. Rh ICP-MS stock-solution (Merck) was diluted to 100 ng/g and 150 µL was added to each sample measured as an internal standard. The Gd-content of the samples was analyzed (*n* = 3 technical replicates) with the aforementioned ICP-MS coupled to an ASX-560 autosampler (Teledyne CETAC, Omaha, NE, USA) with a PFA nebulizer, a cyclonic spray chamber and a 2.5 mm injector. The TQ was again operated with oxygen as a reaction gas and the ions ^158^Gd^16^O^+^ and ^103^Rh^+^ were measured with a dwell time of 100 ms each. Plasma power and argon gas flows were the same as described for the LA-ICP-MS.

### 4.11. Statistical Analysis

Data were tested for normal distribution and a mean value was calculated and presented for all variables. Significance was compared by unpaired and bilateral *t*-test analysis and reported at *p* < 0.05. Statistics were performed using Microsoft Excel (version 16.57; Microsoft, Redmond, WA, USA).

## 5. Limitations

This study has limitations: Most importantly, an orthotopic mouse model would be a more optimal tumor model than the xenograft mouse model, so that the tumor microenvironment can also be considered. In addition, triggering PCa in the mouse prostate would better mimic the pathophysiology of human disease. This requires higher-resolution MRI because the mouse prostate is very small. Molecular characterization is only possible for collagen and shows no general difference in tumor size, which is both a limitation and an advantage. However, the entire progression from the origin to progression or metastasis of collagen would need to be analyzed to form more reliable conclusions.

## 6. Conclusions

Our study shows that MR imaging with a collagen-specific probe is feasible in prostate cancer. Collagen could be visualized in both smaller and larger tumors. However, it is also apparent that the peripheral tumor area expresses more collagen compared to the central area. In the future, this non-invasive method could help to detect tumors and their areas at different stages and identify molecular features. In a heterogenous tumor entity such as prostate cancer, this may help to establish non-invasive imaging biomarkes for diagnosis and disease monitoring, potentially reducing unnecessary biopsies and surgery.

## Figures and Tables

**Figure 1 ijms-24-00711-f001:**
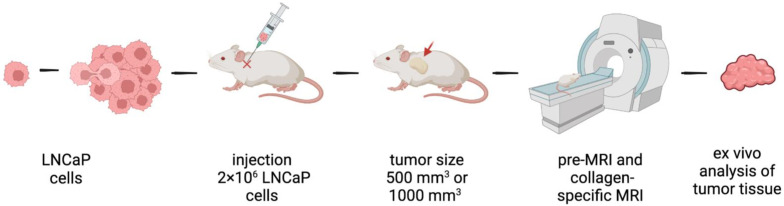
Study design. LNCaP tumor cells were injected subcutaneously into the mouse near the right scapula. After reaching the desired tumor size, a pre-contrast MR image of the tumor was taken, and then the collagen-specific contrast agent was injected intravenously via the tail vein. After the MRI examination, the mouse was euthanized, and the tumor was used for ex vivo experiments.

**Figure 2 ijms-24-00711-f002:**
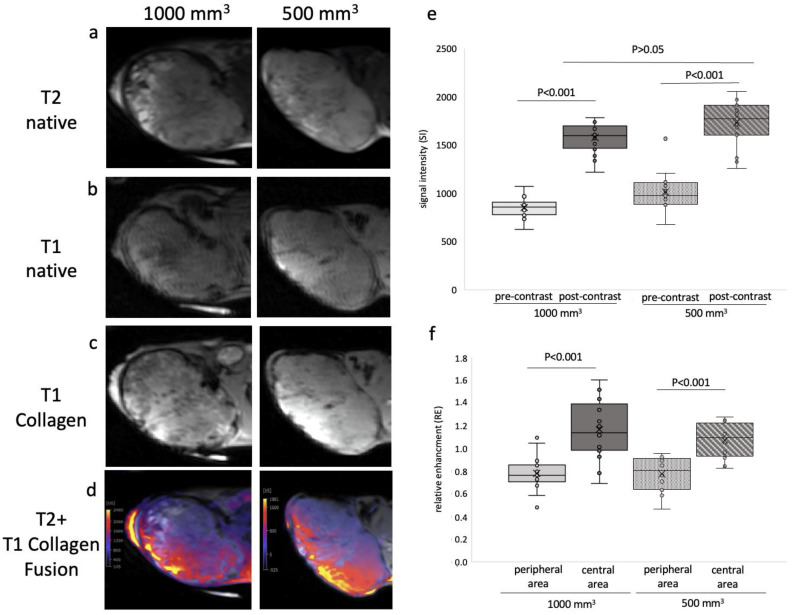
MRI images of prostate cancer in xenograft model of two different tumor sizes, 500 mm^3^ and 1000 mm^3^. (**a**) shows a representative native MRI image of a T2-weighted sequence from a mouse that developed a tumor with a volume of 1000 mm^3^ (**left**) and 500 mm^3^ (**right**) in the scapula area of the mouse. (**b**) shows a native T1-weighted sequence of a tumor-bearing mouse. **Left**: 1000 mm^3^. **Right**: 500 mm^3^. Red arrows show the tumor. (**c**) shows a collagen-specific T1-weighted sequence with the collagen-specific probe of a tumor-bearing mouse. **Left**: 1000 mm^3^. **Right**: 500 mm^3^. Red arrows show the tumor. (**d**) shows a fusion map of the T2-native sequence and T1-probe administration in the same mouse. **Left**: 1000 mm^3^. **Right**: 500 mm^3^. (**e**) demonstrate the analysis of MR images (T1-weighted sequence) before and after collagen-specific administration in two tumor volumes, 500 mm^3^ and 1000 mm^3^. Sixteen mice per group were studied in total (*n* = 16). After administration of the collagen-specific probe, signal intensity (SI) changed in each case (SI of 500 mm^3^: pre-contrast: 916.7 ± 210.6, post-contrast: 1590.9 ± 295.7; SI of 1000 mm^3^: pre-contrast: 800.5 ± 103.6, post-contrast: 1482.0 ± 156.0). The data between pre- and post-contrast were significant (*p* < 0.01). The SI data between post-contrast 500 mm^3^ and post-contrast 1000 mm^3^ was not significant (*p* > 0.05). (**f**) shows the relative enhancement of the peripheral and central area of 1000 mm^3^ (**left**) and 500 mm^3^ (**right**) tumors. The RE data between the peripheral area and central area of 1000 mm^3^ differ significantly (*p* < 0.001, *n* = 16). The RE data between the peripheral area and central area of 500 mm^3^ are significantly different (*p* < 0.001, *n* = 16).

**Figure 3 ijms-24-00711-f003:**
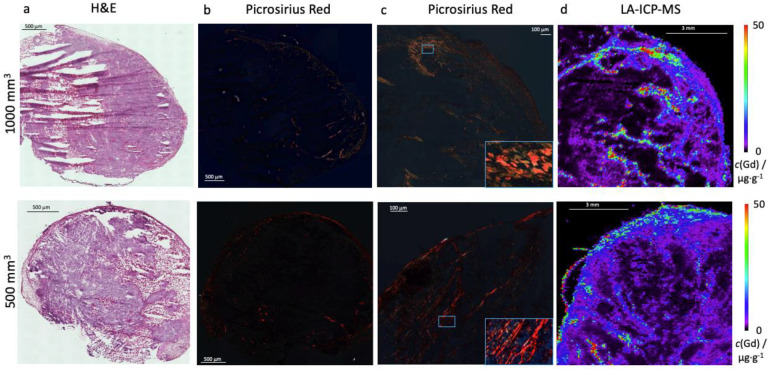
Ex vivo analysis of collagen in 1000 mm^3^ and 500 mm^3^ tumors. (**a**) shows a H&E stain in 1000 mm^3^ tumor (**top**) and 500 mm^3^ (**bottom**). (**b**) shows Picrosirius red staining of collagen of the complete tumor section with a polarization filter in 1000 mm^3^ tumor (**top**) and 500 mm^3^ (**bottom**) and (**c**) shows a representative magnification area. (**d**) LA-ICP MS was performed to localize Gadolinium in the tumor sample. The scale shows the concentration of c(Gd)/µg∙g^−1^.

**Figure 4 ijms-24-00711-f004:**
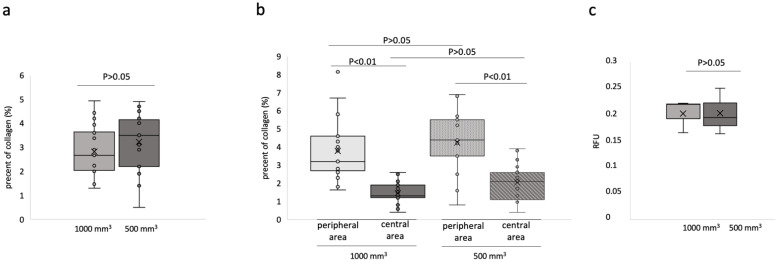
Ex vivo histological and total collagen analyses. (**a**) The percentage of collagen in the histological Picrosirius Red stain was determined. Three areas per slide were calculated and graphically displayed. Per group, *n* = 16 mice were analyzed. The comparison between 500 mm^3^ and 1000 mm^3^ was not significant (*p* > 0.05). (**b**) The histological analysis showed a distinct distribution of collagen fibers. In the peripheral area, more collagen was observed than in the central area. This was the case in both groups (*n* = 16, *p* < 0.001). There was no significant difference in the collagen content of the peripheral areas from 1000 mm^3^ and 500 mm^3^ tumors, respectively (*p* > 0.05). Likewise, the central areas of both tumor sizes demonstrated no significant difference in the amount of collagen fibers detected (*p* > 0.05). (**c**) A total collagen analysis was performed for *n* = 3 tumors per group to calculate the total collagen in the tumor sample for each tumor volume. Both tumor sizes have a similar total collagen content. Only a negligible difference in the collagen content between both groups could be observed (*p* > 0.05).

**Figure 5 ijms-24-00711-f005:**
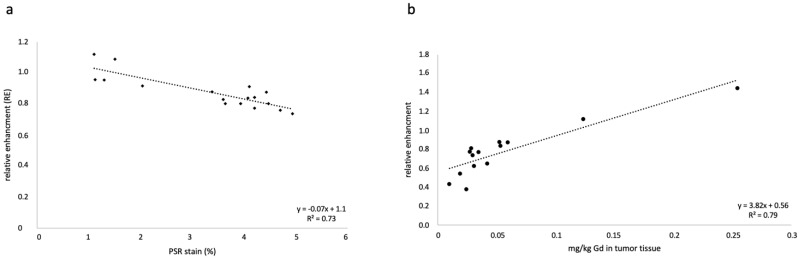
Correlation of the MRI data and the histological and elemental analysis. (**a**) The correlation between the percentage of stained collagen fibers by Picrosirius red (PSR) and the relative enhancements in MRI show a good correlation (y = −0.068x + 1.1; R^2^ = 0.74). (**b**) shows a correlation between the relative enhancement and the ICP-MS analyses for Gd. A good correlation is shown (y = 3.82x + 0.56; R^2^ = 0.79).

## Data Availability

Correspondence can be requested for research data.

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
