# Peer review of "Collagen-Specific Molecular Magnetic Resonance Imaging of Prostate Cancer"

_ijms, 2022, doi:10.3390/ijms24010711_

Round 1
Reviewer 1 Report
1. In this study, the authors used a collagen-specific NMR probe, but there was no structural diagram or related characterization of the probe in the paper.
2. The tumor section staining diagram shown in Figure 3 is partial, please give the complete picture.
3. The data distribution in Figure 5b is not very uniform, and the number of samples with high signal and low signal should be balanced.
4. More details should be noted. Such as subscripts, superscripts in manuscript and reference, and also the format of the references needs to be unified.

Author Response
- In this study, the authors used a collagen-specific NMR probe, but there was no structural diagram or related characterization of the probe in the paper.
The collagen-specific probe was initially developed by Caravan et al., so we did not include the structural diagram in the manuscript. But we have included a note in the manuscript referring to the developers and adding characteristics of the probe.
S.9, line: 298-300
„For this study, a collagen-specific probe, EP-3533 [28, 51] was used. The sample was developed by Caravan et al. [28]. It is a cyclic peptide that binds specifically against collagen type 1.“
- The tumor section staining diagram shown in Figure 3is partial, please give the complete picture.
We thank the reviewer for this comment. We have modified the figure.
- The data distribution in Figure5bis not very uniform, and the number of samples with high signal and low signal should be balanced.
We thank the reviewer for this comment. Due to the availability of the samples, we are limited to performing further measurements. In the case of the high signal, it is possible that we measured a piece of tumor with particularly high Gd. This could be the reason for the variance.
The manuscript has the following statement:
“For the determination of the total Gadolinium (Gd) concentration in the tumor samples a cross-section of the tumor probe was prepared (n=7 per group) and then dehydrated under a vacuum atmosphere (vacuum pumping device, vacuubrand, Wertheim, Germany).” (S.10, line: 284-287)
- More details should be noted. Such as subscripts, superscripts in manuscript and reference, and also the format of the references needs to be unified.
We thank the reviewer for this comment. We have modified the manuscript.

Reviewer 2 Report
Collagen fibers are responsible for kinds of cell activity and character, such as tissue stiffness, migration, invasion/metastasis, and so on. With a collagen-specific probe, the author explores the content of this molecular marker for two mouse models with different tumor volumes. A significant difference between the signal intensity of the peripheral tumor area and the central area of the tumor was found, compared with the similarity of the collagen content for the two tumor volumes, and this may help to differentiate the tumor from healthy tissue. Before it can be accepted, some issues need to be concerned.
1. In the manuscript, the volume of the tumor tissue was selected as 500 and 1000 to do the MRI detection for the collagen. Is there any special reason for these two volumes in the development of the tumor?
2. How to define the peripheral area and the central area, or define the boundary of these two areas?
Author Response
- In the manuscript, the volume of the tumor tissue was selected as 500 and 1000 to do the MRI detection for the collagen. Is there any special reason for these two volumes in the development of the tumor?
In previous studies we demonstrated that these two sizes could already show differences in the extracellular matrix components. For example, the smaller tumors, 500 mm3, showed a higher signal intensity with the elastin-specific probe than the 1000 mm3 tumors. The ex vivo studies confirmed the in vivo data (Kader et al. 2021). In another study, we investigated iron-oxide nanoparticles in different tumor sizes, 500 mm3 and 1000 mm3. Here we were shown that a higher signal reduction was seen in the smaller tumors on MRI. We were able to confirm the in vivo results in the histological examinations.
Based on these observations, we decided to use these sizes.
We have added a line in the manuscript.
s.8, line: 284-286
“In previous studies, it was shown that differences in the composition of the ECM were visible at the sizes of 500 mm3 and 1000 mm3, so these two sizes were chosen.”
Kader A, Brangsch J, Reimann C, et al. Visualization and Quantification of the Extracellular Matrix in Prostate Cancer Using an Elastin Specific Molecular Probe. Biology. 2021 Nov;10(11):1217. DOI: 10.3390/biology10111217. PMID: 34827210; PMCID: PMC8615039.
Kader A, Kaufmann JO, Mangarova DB, et al. Iron Oxide Nanoparticles for Visualization of Prostate Cancer in MRI. Cancers (Basel). 2022 Jun 13;14(12):2909. DOI: 10.3390/cancers14122909. PMID: 35740575; PMCID: PMC9221397.
- How to define the peripheral area and the central area, or define the boundary of these two areas?
We thank the reviewer for this comment. We added a paragraph in the manuscript.
S.9 , line: 318-321
“The central area was defined by means of a midpoint. Starting from this point, a radius was defined, which was measured in each case. For the peripheral area, an outermost point starting from the midpoint was selected, that was measured in each case.”
